# Vitamin D3 in High-Quality Cow Milk: An Italian Case Study

**DOI:** 10.3390/foods9050548

**Published:** 2020-05-01

**Authors:** Mara Mandrioli, Emanuele Boselli, Federica Fiori, Maria Teresa Rodriguez-Estrada

**Affiliations:** 1Department of Agricultural and Food Sciences, Alma Mater Studiorum—Università di Bologna, Viale Fanin 40, 40127 Bologna, Italy; mara.mandrioli@unibo.it (M.M.); maria.rodriguez@unibo.it (M.T.R.-E.); 2Free University of Bolzano, Faculty of Science and Technology, Piazza Università 1, 39100 Bozen-Bolzano, Italy; 3Department of Agricultural, Food and Environmental Sciences (D3A)—Università Politecnica delle Marche, Via Brecce Bianche, 60131 Ancona, Italy; federica.fiori@hotmail.it; 4Interdepartmental Centre for Industrial Agrofood Research, Alma Mater Studiorum, University of Bologna, Via Quinto Bucci 33, 47521 Cesena, Italy

**Keywords:** cholecalciferol, high-quality cow milk, Italy, pasteurized cow milk, raw cow milk, vitamin D3, consumers, high-performance liquid chromatography (HPLC), mass spectrometry

## Abstract

The quality-labeling category of high-quality (HQ) milk defined by the Italian legislation must comply with specific requirements concerning rigorous breeder management, hygienic controls, fat and protein content, bacterial load, somatic cells, lactic acid content, and non-denatured soluble serum proteins. However, there is no specification for the vitamin D content of HQ milk. Moreover, the data on the vitamin D content of this milk category are very scarce. In the present study, the content of vitamin D3 was evaluated in HQ raw and pasteurized cow milk obtained from Italian cowsheds and supermarkets. The vitamin D3 content varied from not detected (less than 1 µg L^−1^) to 17.0 ± 2.0 µg L^−1^ milk and was not related to the milk fat content. These results represent a case study including a significant although not exhaustive part of the contemporary Italian market of HQ milk. It was shown for the first time that HQ raw milk does not necessarily contain more vitamin D3, even though non-expert consumers likely to buy milk labeled as HQ could expect it. The vitamin D3 content in HQ pasteurized whole milk should be reported on the label of the milk package as a best practice of consumer information policy.

## 1. Introduction

Vitamin D is a fat-soluble vitamin that plays an essential role in bone metabolism, by properly adjusting the concentration of calcium and phosphorus in the human body. Many studies have shown the correlation between the dietary intake of vitamin D and improvement in certain diseases, such as glycemic index in patients with diabetes mellitus type I [1]. Hypovitaminosis D has also been associated with loss of muscle mass and bone frailty, especially in elderly people [2]. Very recent studies suggest that proper supplementation of vitamin D3 may enhance our resistance to SARS-CoV-2 [3], influenza and pneumonia infections [3,4], as well as may reduce the incidence and severity of COVID-19 by increasing the serum 25-hydroxyvitamin D concentration [5].

Vitamin D is synthesized in humans, mainly by skin exposure to UV-B sunlight. The two major forms are D2 (ergocalciferol) and D3 (cholecalciferol) from plant and animal food, respectively. The major food sources of vitamin D are fishery products, eggs, cereals and dairy products.

Vitamin D3 is present in milk and diverse concentrations have been reported: 0.13–1.0 µg/L [6], 0.13–0.88 µg/L [7], 0.08–1.35 µg/kg and 0.03–1.86 µg/kg in whole milk and whole organic milk, respectively [8]. Nevertheless, milk has always been considered a poor source of vitamin D.

The content of fat-soluble vitamins A, D, and E in milk is strongly influenced by the food ration composition of the dairy cow [9]. The vitamin D content is greatly affected by the lipid content and season. Moreover, some differences have been observed between organic and conventional milk [8]. Currently, the interest and attention of livestock production are aimed at improving not only human nutrition [10] but also the cattle’s health conditions. The technique of constant or ‘unifeed’ ration (from unique fed, or “unique food”) is a cattle feeding method, which has replaced the traditional method (or ‘‘single dish’’) of feed administration. The efficacy of the composition and quality of the food ration ‘unifeed’ has been widely demonstrated; thus, this type of food ration is widely used in the livestock production to improve the nutritional profile of milk.

In Italy, the quality-labeling category of high-quality (HQ) milk is regulated by the Decree of the Italian Ministry of Health no. 185/1991 [11], which imposes rigorous breeder management and hygienic controls regarding fat and protein content, bacterial load, somatic cells, lactic acid content, and non-denatured soluble serum proteins [12]. Although there is no indication about the vitamin D content of HQ milk, consumers would presumably expect a higher content of vitamin D in this milk category, since the starting HQ raw milk must comply with more precise and stricter hygienic and composition requirements. HQ milk has a higher minimum fat content and must be subjected to a gentler pasteurization treatment, such as high temperature-short time pasteurization (HTST, 72 °C for 15 s) which is supposed to better preserve the micronutrient content (fat-soluble vitamins included) with respect to conventional pasteurization.

The assessment of vitamin D in milk is a laborious analytical determination, due to the matrix complexity and the need for high sensitivity. The presence of other minor lipophilic components (such as carotenoids, retinol, tocopherols, and sterols) can interfere with vitamin D determination, causing an overestimation. In addition, the sample purification step can lead to large analyte losses if the extraction yield is not optimal. For these reasons, it is necessary to apply analytical methods with high specificity. The most widely used analytical procedure involves saponification, purification, and quantification with high-performance liquid chromatography (HPLC) coupled to the ultraviolet-diode array detector (UV–DAD) or mass spectrometry (MS) detector [13] to confirm the identification.

Since the concentration of cholecalciferol in milk is important to assess the daily intake in humans, the present study aimed at evaluating the content of vitamin D3 in raw milk and pasteurized HQ whole milk obtained according to the Italian Ministerial Decree 185/1991 [11]. The obtained data set presented here for the first time on HQ milk represents a case study including a significant although not exhaustive part of the contemporary Italian market for this specific quality-labeling category of milk.

## 2. Materials and Methods

### 2.1. Chemicals

All chemicals and solvents were of analytical grade. HPLC-grade methanol (≥99.9%), ammonia 14 mol/L, ethanol (96%), ethyl ether (≥99.5%), *n*-pentane, aqueous solution of Na_2_SO_4_ at 10% (*w*/*v*), anhydrous sodium sulfate powder (≥99.9%), and water (≥ 99.8%), were purchased from Merck (Darmstadt, Germany). The standards of vitamins D2 and D3 were supplied by Sigma (St. Louis, MO, USA). Silica solid-phase extraction (SPE) cartridges (Strata SI-1, 55 µm, 70 Å, 500 mg/3 mL) from Phenomenex (Torrence, CA, USA), were used for vitamin D3 purification.

### 2.2. Sampling

Twenty-two samples of HQ raw milk (FARM) were collected directly from farms producing HQ milk (according to the guidelines of the Italian Ministerial Decree 185/1991), which delivered it directly to an Italian dairy company. The farms were mainly located in Emilia Romagna, Lombardia and Veneto regions, and were of medium and large size. The Friesian dairy cows were fed with unifeed or “single pot”, which could guarantee the administration of a homogeneous and nutritionally balanced food ration. To maintain health and animal welfare, the daily food ration was added with a “mineral feed” for the integration of vitamins and oligo elements. Given that the supplement intake with the daily ration was about 500 g per day per head, and that it represented the only source of vitamin D3 intake, a daily intake per head of 950 µg of vitamin D3 could be assumed. Milk sampling was carried out by collecting raw mass milk of at least 2 or 4 milkings. Before sampling, the homogenizer was operated for few min, in order to avoid fat creaming at the milk surface and thus obtain a representative sample of the entire milk mass. Milk was collected in sterile 1.5 L PET bottles, kept at a temperature between 2 and 6 °C during transportation, and frozen at −18 °C until analysis.

Eight samples of pasteurized HQ whole milk (PHQ) from the same dairy company were purchased in an Italian supermarket in the same period. To obtain PHQ, milk underwent homogenization and HTST pasteurization at 72–75 °C for 15 s.

All FARM and PHQ samples were subjected to fat cold extraction according to the procedure described in Section 2.4, as well as to the determination and quantification of the vitamin D3 content by high performance liquid chromatography coupled to a photodiode array (HPLC/UV-DAD) detector and HPLC coupled to mass spectrometry (HPLC/MS). Two independent replicates were performed for each sample.

### 2.3. Analytical Methods

Vitamin D3 was determined with an analytical procedure consisting of direct cold saponification, extraction of the unsaponifiable fraction [14], purification of the unsaponifiable matter by silica solid phase extraction (SPE) [15], and injection into a HPLC system equipped with a UV-DAD detector and a MS detector [16,17]. Quantification of cholecalciferol was performed by using vitamin D2 as internal standard (IS).

#### 2.3.1. Direct Cold Saponification

An aliquot of 35 g of milk was placed in a 100-mL glass bottle, added with 2.36 μg of vitamin D2 as IS, 35 mL of a 35% ethanolic solution of KOH (in ethanol 85%) and 20 mL of an ethanolic solution of 1% pyrogallol. The bottle headspace was then flushed with nitrogen to remove oxygen. Afterwards, the bottle was agitated in the dark for 18 h to allow complete saponification. Successively, the unsaponifiable fraction was extracted twice with 35 mL of a mixture of petroleum ether:diethyl ether (90:10, *v*/*v*), as suggested in [14]. The ethereal phase was washed 3 times with 30 mL of water until a neutral pH was reached. The ether extract was dried by adding 10 g of anhydrous sodium sulfate, for a total contact time of 60 min at 4 °C. After filtration with a paper filter, the ethereal phase was dried in a rotary evaporator. The unsaponifiable matter was dissolved in 1 mL of *n*-hexane and stored at −20 °C until analysis.

#### 2.3.2. Solid Phase Extraction (SPE) of Unsaponifiable Fraction

For the determination of vitamin D3, the unsaponifiable matter was purified using silica SPE as described in [12]. SPE cartridges packed with 500 mg of silica, were used. After cartridge activation with *n*-hexane (5 mL), the phase was washed with 2 mL of *n*-hexane:chloroform (22:78, *v*/*v*) and vitamin D3 was finally eluted with methanol (2 mL). The methanolic extract was dried under gentle nitrogen flow and the sample was dissolved into a smaller volume of the same solvent.

#### 2.3.3. Chromatographic Determination of Vitamin D3

The sample was analyzed by HPLC coupled on-line with a UV-DAD detector and a MS detector (to confirm the identification of vitamin D3) using the method suggested by Sliva et al. [16] and Hymøller and Jensen [17], with slight modifications. The HPLC system consisted of a quaternary pump Varian ProStar 240 (Palo Alto, CA, USA) and a 50-µL loop. The instrument was coupled to an LCQ-Duo (Thermo Finnigan, San Jose, CA, USA), equipped with an electrospray interface (ESI). Vitamin D2 (internal standard) and vitamin D3 were identified with positive ionization (ESI^+^) as their pseudomolecular ions 397.65 *m/z* and 385.01 *m/z*, respectively. For quantitative analysis, a photodiode array (PDA) detector mod. 330 (Varian, Palo Alto, CA, USA) equipped with an analytical YMC30 column (250 mm × 4.6 mm i.d., 5 µm particle size) (YMC Separation Technology, Dinslaken, Germany) packed with a C30 stationary phase, was used. The guard column (10 mm × 4.0 mm i.d.) was packed with the same stationary phase. Samples were injected in an isocratic mode, with a methanol:water (96:4, *v*/*v*) mobile phase, at a flow rate of 1 mL/min. The injection volume was 50 μL. The wavelength of acquisition was 265 nm and the temperature of both columns was held at 50 °C into a thermostated heater. The detection limit (LOD) was 1 µg L^−1^ milk, while the quantification limit (LOQ) was 3 µg L^−1^ milk. LOD and LOQ were calculated as signal-to-noise ratios equal to 3:1 and 10:1, respectively.

### 2.4. Fat Extraction

The extraction of milk lipids was performed according to the ISO 14156-2001 IDF method [18]. One hundred mL of milk were placed in a 500-mL separating funnel, to which 80 mL of ethanol, 20 mL of ammonia and 100 mL of ethyl ether were added. The funnel was shaken vigorously for 1 min and, when the phases were separated, 100 mL of pentane were added and again stirred gently. Once phase separation was achieved, the aqueous fraction was eliminated and the organic phase washed twice with 100 mL of 10% Na_2_SO_4_ solution. The organic phase was then dried with anhydrous sodium sulfate for 10 min; after being filtered, the solvent was first evaporated with a rotary evaporator at 40 °C and thereafter with a slight nitrogen flow. The extracted fat was placed in an oven at 50 °C until a constant weight was attained, and it was then weighed. Two replicates for each sample were performed.

### 2.5. Statistical Analysis

The data of vitamin D3 and fat content are reported as average values ± standard deviation (St) of two independent replicates (biological replicates). Pearson correlation coefficients between fat and vitamin D3 content were calculated using Microsoft Excel software (version 2010).

## 3. Results

### 3.1. Vitamin D3 Content

The HPLC/UV-DAD chromatogram of the unsaponifiable fraction of HQ milk registered at a wavelength of 265 nm [16], is reported in Figure 1. Several components, such as fat-soluble vitamins, sterols and hydrocarbons, make up the unsaponifiable matter of milk, leading to a complex biological mixture. This complexity is clearly shown by the rich chromatographic trace. For this reason, the identification of the analytes of interest was performed with an integrated approach using both a spiking procedure and the identification of vitamin D3 through HPLC/MS.

To identify vitamin D3 (and vitamin D2, if present), the milk sample was spiked with the pure compounds (central and bottom panel of Figure 1).

It was possible to exclude the presence of vitamin D2 in all the evaluated samples using the spiking procedure. This can be easily verified by comparing the top and central chromatograms of Figure 2, in which the peak of the spiked vitamin D2 is indicated as “b”. Moreover, vitamin D3, which was naturally present in the milk sample (peak “a” in the top chromatogram), eluted about one minute later than vitamin D2 under the analytical conditions used, thus assuring no peak overlapping.

As a second necessary step, vitamin D3 (the “a” peak in the top chromatogram of Figure 1) was unambiguously identified using HPLC/tandem MS (MS^2^) under the same HPLC conditions. The mass fragmentation of the pseudomolecular peak of vitamin D3 is reported in Figure 2. The dissociation of the pseudomolecular peak at *m/z* 385.01 generates fragments related to the loss of water (385 − 18 = 367), and the contemporary loss of water and the isopropyl side chain (385 − 18 − 42 = 325).

The use of an IS instead of an external calibration was already reported in an interlaboratory study by Staffas and Nyman [19]. Basically, the sample purification for vitamin D3 is a laborious protocol (sample saponification followed by solvent extraction and SPE), thus the loss of the analyte during the procedure cannot be considered constant and a correction factor cannot be determined. The IS method was chosen for a simple but crucial reason: since the IS was added directly in the milk (at a very early stage), losses along the various steps of the purification procedure were automatically compensated for [20]. In addition, the choice of this analytical approach is further supported by the following facts: (i) vitamin D2 (IS) is easily distinguished from the analyte in the HPLC trace, as reported in Figure 1; (ii) vitamin D2 exhibits similar chemical properties to vitamin D3, so it is assumed that they undergo a similar loss during sample pretreatment; (iii) the presence of vitamin D2 was excluded in all the evaluated samples; (iv) the extinction coefficient E (1%, 1 cm) at 265 nm is extremely similar for both vitamins D2 and D3 (475 and 480, respectively) [19].

Known quantities of ergocalciferol (at three different concentration levels (0.01, 0.03 and 0.06 mg kg^−1^)), were added to calculate the vitamin D2 recovery after sample saponification and purification. The average extraction yield was equal to 81.9%, which is within the expected recovery ranges (60–115%) at the tested concentrations, according to the Procedural Manual of the Codex Alimentarius [21]. In addition, the obtained extraction recovery agrees with those reported in other studies, where vitamin D3 in milk was determined with similar analytical methodologies [13,22]. As a result, vitamin D2 was successfully used as IS for the quantification of vitamin D3 in all the samples.

Table 1 reports the vitamin D3 content expressed as µg L^−1^ of milk. In PHQ milk, the vitamin D3 content varied from not detected (<1 µg L^−1^) to 17.0 µg L^−1^. Only four samples evidenced the clear presence of vitamin D3, while, in the other samples, cholecalciferol was found at trace levels (<3 µg L^−1^), except for one sample where it was absent. In FARM samples, the vitamin D3 content ranged from not detected to 12.5 µg L^−1^. In this case, only three samples had quantifiable values of vitamin D3, whereas in most samples it was either not detected (*n* = 10) or present only at trace levels (<3 µg L^−1^, *n* = 9). In the raw milk samples (FARM1 and FARM2), vitamin D3 was below the quantification limit (<3 µg L^−1^).

### 3.2. Fat Content

Table 1 also reports the fat content of FARM and PHQ samples and expressed as g 100 mL^−1^ milk (%). According to the ISO 14156:2001 method, the fat content of PHQ samples ranged from 2.84% to 3.21%, while in FARM milk it varied from 2.55% to 4.23%. It is not surprising that the fat content is lower than the legal requirement defined by the Italian regulation. It has already been reported [23] that, although this is the official International Standardization Organization-International Dairy Federation method for milk lipid extraction, it gives satisfactory results for neutral lipids, but it has important disadvantages in terms of polar lipid losses, such as phospholipids and sphingolipids. Thus, it can be assumed that these results are a good approximation of the content of neutral lipids in milk, which also include the unsaponifiable matter containing vitamin D3, since this is a non-polar fraction of the total lipids.

## 4. Discussion

The results obtained in the present study agree with those reported in the literature for non-HQ milk, which show that the vitamin D3 levels in non-enriched fresh bovine milk, including HQ milk, are generally very low. Trenerry et al. [24] detected 0.05 µg 100 mL^−1^ of vitamin D3 in milk (~4.5% fat), while Perales et al. [6] found 0.125-1.0 µg L^−1^ of vitamin D3 in milk.

In commercial pasteurized milk, vitamin D3 levels were correlated with the fat content in a previous research [8]. However, in the present study, no correlation between the vitamin D3 content and the fat content measured with the ISO method could be deduced from the data reported in Table 1. In other words, many of the analyzed samples had a vitamin D3 level below the quantitation limit (<3 µg L^−1^), despite the relatively high milk fat content, and vice versa. This result was not affected by the fat extraction method, because vitamin D3 is a non-polar lipid component, for which the ISO method represents the optimal extraction procedure. In any case, these results also reflect the complexity of the cow physiology, underlying the fact that not all the feed ingredients are bio-accumulated in milk, as many of these components are actually used in the cow metabolism (bones, muscles, cardiovascular apparatus, etc.).

In general, vitamin D3 was often more abundant in PHQ samples than in FARM ones; this could be due to the dairy industry practice of mixing the collected HQ raw mass milk from the diverse farms to obtain the required fat content without adding any cream or skim milk, as ruled by the Italian Decree no. 185/1991 [11]. Considering that raw milk needs to be boiled before consumption according to the current Italian regulation, this might further reduce vitamin D3 content due to its possible thermal degradation and consequent reversible isomerization to pre-vitamin D3 [25]. However, thermal stability of vitamin D is still a controversial aspect, as other studies have proved that vitamin D is heat stable and that pasteurization and sterilization treatments do not promote its degradation [26,27].

On the other hand, the guidelines for healthful Italian food habits published by the Italian National Institute for Research on Food and Nutrition [28] recommend an average consumption of three 125-mL portions of milk per day, considering an average content of 9 µg/kg of vitamin D3 in HQ pasteurized whole milk (equivalent to 135 UI of vitamin D3).

## 5. Conclusions

This work presents the vitamin D3 content of HQ milk (a quality-labeling milk category defined by the Italian legislation) for the first time in a scientific publication. It is a survey about the differences in vitamin D3 content between HQ raw and pasteurized milk with the aim to monitor a representative even if not exhaustive part of the Italian production (particularly from Northern Italy). Although the HQ regulation does not provide specific requirements for vitamin D3 in HQ milk, the consumer could expect that HQ milk contains more vitamin D3 than conventional milk, since HQ raw milk must comply with more precise and stricter hygienic and composition requirements, especially in terms of fat content. However, in general, the content of vitamin D3 in HQ milk was not higher than conventional milk. In some cases, the vitamin D content of pasteurized HQ milk was higher than raw HQ milk.

Since the daily requirement of vitamin D for healthy adults is 600 IU [29] in a balanced diet, it would be highly advisable and useful that the actual content of vitamin D3 in HQ pasteurized whole milk were added as nutritional information on the label of the milk package, as a best practice of consumer information policy.

## Figures and Tables

**Figure 1 foods-09-00548-f001:**
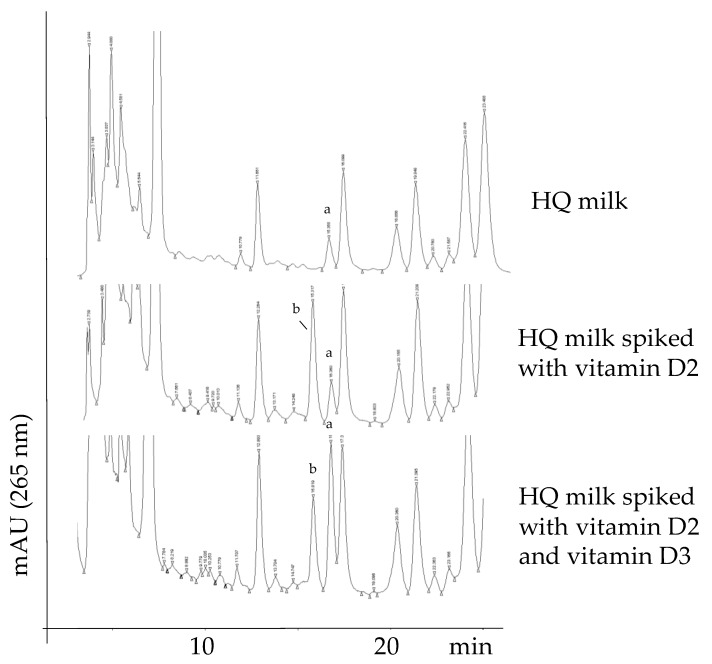
HPLC/UV-DAD traces of a high-quality (HQ) milk sample showing the natural presence of vitamin D3 (a) and absence of vitamin D2 (top panel); the same HQ milk sample spiked with vitamin D2 (b) (central panel) and spiked with both vitamin D2 (b) and vitamin D3 (a) (bottom panel). Peak identification: a, vitamin D3 (naturally present or spiked); b, vitamin D2 (present only in the spiked samples).

**Figure 2 foods-09-00548-f002:**
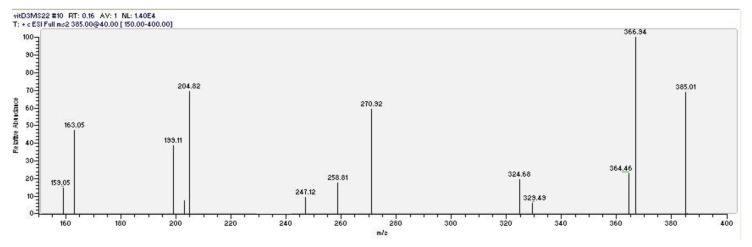
Mass fragmentation of the pseudomolecular ion (M + 1 = 385)^+^ of vitamin D3.

**Table 1 foods-09-00548-t001:** Content of vitamin D3 (µg L^−1^ of milk) and fat (according to ISO 14156:2001 method) in raw HQ milk (FARM) and pasteurized HQ milk (PHQ) samples (± standard deviation). Abbreviation: nd, not detected (<1 µg L^−1^ milk).

Sampling	Vitamin D3 (µg L^−1^ Milk)	Fat Content (g 100 mL^−1^ Milk)
Raw HQ cow milk		
FARM 1	<3	3.53 ± 0.03
FARM 2	nd	3.26 ± 0.02
FARM 3	<3	4.23 ± 0.34
FARM 4	<3	3.07 ± 0.02
FARM 5	<3	2.61 ± 0.06
FARM 6	<3	3.24 ± 0.00
FARM 7	<3	2.92 ± 0.05
FARM 8	7.9 ± 0.9	2.74 ± 0.00
FARM 9	12.5 ± 2.1	3.19 ± 0.02
FARM 10	5.4 ± 0.6	3.84 ± 0.06
FARM 11	<3	2.93 ± 0.01
FARM 12	<3	2.99 ± 0.00
FARM 13	nd	3.49 ± 0.05
FARM 14	nd	3.35 ± 0.23
FARM 15	nd	3.38 ± 0.08
FARM 16	nd	3.40 ± 0.25
FARM 17	nd	3.78 ± 0.22
FARM 18	nd	3.67 ± 0.18
FARM 19	<3	3.47 ± 0.06
FARM 20	nd	2.55 ± 0.35
FARM 21	nd	3.89 ± 0.12
FARM 22	nd	4.06 ± 0.11
Pasteurized HQ cow milk		
PHQ 1	15.0 ± 1.3	3.15 ± 0.01
PHQ 2	11.0 ± 1.2	2.84 ± 0.02
PHQ 3	17.0 ± 2.0	3.07 ± 0.03
PHQ 4	<3	3.02 ± 0.02
PHQ 5	nd	3.06 ± 0.03
PHQ 6	<3	2.93 ± 0.02
PHQ 7	9.3 ± 1.0	3.16 ± 0.02
PHQ 8	<3	3.21 ± 0.07

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
