# Peer review of "Vitamin D3 in High-Quality Cow Milk: An Italian Case Study"

_foods, 2020, doi:10.3390/foods9050548_

Round 1

Reviewer 1 Report

This manuscript is clearly written and designed to evaluate VD3 status in high quality milk. This is useful information that improves our understanding of VD3 availability in the high quality milk. The rationale, experimental design, methods, data presentation, and discussion is all presented in a logical, easy to understand manner.

Please check the manuscript for typing and spelling errors.

Author Response

The reviewer says:

'Please check the manuscript for typing and spelling errors.'

We will double check the manuscript for typing and spelling errors accurately. We thank the reviewer very much for this suggestion.

Reviewer 2 Report

The study presents the content of vitamin D3 and fat in high-quality cow milk in Italia. Milk is a very complex matrix. The analysis of vitamin D3 in milk is not easy. Nevertheless, the authors have well analyzed the content of vitamin D3.

To quantify the vitamin D3, you used vitamin D2 as an internal standard instead of standard of vitamin D3 for external curve. Do you have any reason?

What is the injection volume for HPLC coupled with a PDA detector and a mass detector? Please mention it in Method section.

You got a 81.9% extraction recovery using vitamin D2. Is it an acceptable range for recovery%? Please compare it and mention in the section.

Author Response

Point 1: The study presents the content of vitamin D3 and fat in high-quality cow milk in Italia. Milk is a very complex matrix. The analysis of vitamin D3 in milk is not easy. Nevertheless, the authors have well analyzed the content of vitamin D3.

We thank the reviewer very much for the very positive comment on our paper.

Point 2: To quantify the vitamin D3, you used vitamin D2 as an internal standard instead of standard of vitamin D3 for external curve. Do you have any reason?

The use of an internal standard instead of an external calibration was already reported in an interlaboratory study by Staffas and Nyman (JAOAC, 86,2,2003) (this citation was added). In addition, we based our approach on the Guidelines For Dietary Supplements and Botanicals of the AOAC Official Methods of Analysis (2013), Appendix K, p. 7 (freely available in the internet; we also added this citation).

Basically, the sample purification for Vit D3 is a laborious protocol (sample saponification, solvent extraction, SPE), thus the loss of analyte during the procedure cannot be considered constant and a correction factor cannot be determined, as reported in the cited Guidelines for the External Standard Method. The use of the Internal Standard Method was chosen for a simple but crucial reason: since the Internal Standard was added directly in the milk (at a very early stage), losses along the various steps of the purification procedure were automatically compensated for.

In addition:

1) vitamin D2 (the internal standard, IS) is easily distinguished from the analyte in HPLC, as reported in Figure 1, but exhibits similar chemical properties, so it is assumed that it undergoes a similar loss during sample pretreatment.

2) it was possible to exclude the presence of vitamin D2 in all the evaluated samples (as already reported in the manuscript)

3) The extinction coefficient E (1%, 1 cm) at 265 nm is extremely similar for both vit. D2 and D3 (475 and 480, respectively) (Staffas and Nyman, 2003)

The manuscript has been added with all this information in the section 3. Results.

Point 3: What is the injection volume for HPLC coupled with a PDA detector and a mass detector? Please mention it in Method section.

The injection volume was 50 uL. We clarified this in the section 2.3.3 (Chromatographic determination of vitamin D3).

Point 4: You got a 81.9% extraction recovery using vitamin D2. Is it an acceptable range for recovery%? Please compare it and mention in the section.

Yes, an acceptable recovery is a function of the concentration. According to the Procedural Manual of the Codex Alimentarius Commission, 27th Ed. (Joint FAO/WHO Food Standards Program 2019), the expected recovery ranges at the tested concentrations is 60–115%; thus, 81.9% fully meets this parameter. In addition, the extraction recovery here obtained agrees with those reported in other studies, where they determined the content of vitamin D3 in milk with similar analytical methodologies (Kasalovà et al., 2015; Sereshti et al., 2020).

We added these comments and the missing citations in the section Results.